# Molecular and Biological Characterization of an Isolate of Fusarium graminearum dsRNA mycovirus 4 (FgV4) from a New Host *Fusarium pseudograminearum*

**DOI:** 10.3390/microorganisms13020418

**Published:** 2025-02-14

**Authors:** Guoping Ma, Yueli Zhang, Liguo Ma, Kai Cui, Bo Zhang, Hang Jiang, Kai Qi, Junshan Qi

**Affiliations:** 1Shandong Key Laboratory for Green Prevention and Control of Agricultural Pests, Institute of Plant Protection, Shandong Academy of Agricultural Sciences, Jinan 250100, China; guopingmasaas@163.com (G.M.); zhangyuelisaas@163.com (Y.Z.); maliguosaas@163.com (L.M.); zhangbosaas@163.com (B.Z.); jianghangsaas@163.com (H.J.); 2Shandong Provincial Key Laboratory of Test Technology on Food Quality and Safety, Institute of Quality Standard and Testing Technology for Agro-Products, Shandong Academy of Agricultural Sciences, Jinan 250100, China; kaicuisaas@163.com

**Keywords:** *Fusarium pseudograminearum*, mycovirus, orthocurvulavirus, biological characteristics, biocontrol

## Abstract

Wheat Fusarium crown rot (FCR), mainly caused by *Fusarium pseudograminearum*, is one of the most important diseases. Some mycoviruses are reported to have a hypovirulence trait and considered as a biocontrol agent for plant fungal diseases. In most cases, mycovirus biological effects have not been explored clearly. In this study, we identified and characterized a novel isolate of double-stranded RNA (dsRNA) mycovirus, Fusarium graminearum dsRNA mycovirus 4 (FgV4), from a new host, an isolate WC9-2 of *F. pseudograminearum*. The genome of FgV4-WC9-2 includes two dsRNA segments of 2194 bp and 1738 bp. FgV4-WC9-2 dsRNA1 contains a single open reading frame (ORF1), which encodes a protein of 675 amino acids (aa) and has a conserved RNA-dependent RNA polymerase (RdRp) domain. FgV4-WC9-2 dsRNA2 contains two discontinuous ORFs (ORF2-1 and ORF2-2) that code for hypothetical proteins with unknown function. Biological characteristics research has shown that FgV4-WC9-2 infection did not change the colony morphology, but it could significantly decrease colony growth rate. FgV4-WC9-2 could also reduce the sporulation ability, change the conidia size and reduce the pathogenicity of the host to a certain extent. This study is the first to describe a hypovirulence-associated orthocurvulavirus infecting *F. pseudograminearum*, which has the potential to assist with FCR disease biological management.

## 1. Introduction

Fusarium crown rot (FCR) of wheat (*Triticum aestivum* L.) has become one of the most destructive soil-borne or residue-borne diseases in most wheat planting regions over the world [1]. FCR occurs throughout the growth period of wheat. FCR can cause seedling death before or after emergence. The remaining seedlings may develop typical disease symptoms, including coleoptile, subcrown internode, sheaths of lower leaf, stems and internode browning in the lower part of the plant, and white heads with shriveled or no grains in severe cases [1]. Many *Fusarium* species are reported to be pathogens of wheat FCR disease, such as *Fusarium acuminatum*, *F. asiaticum*, *F. avenaceum*, *F. babinda*, *F. crookwellense*, *F. culmorum*, *F. equiseti*, *F. flocciferum*, *F. graminearum*, *F. incarnatum*, *F. ipomoeae*, *F. oxysporum*, *F. proliferatum*, *F. pseudograminearum*, *F. sinensis*, *F. subglutinans*, *F. torulosum*, and *F. tricinctum* [2,3,4,5]. Among these *Fusarium* species, *F. pseudograminearum* has been proven to be the predominant species, and can cause large yield and economic losses [3,4,5].

Mycoviruses (fungal viruses) are viruses that infect and replicate in fungal cells. Mycoviruses are found in all major fungal taxa [6]. The first mycovirus was found in a diseased mushroom (*Agaricus bisporus*) in 1962; the mycovirus-infected mushrooms displayed slow growth, fruiting body malformation, and early maturation, causing serious yield losses [7]. Mycovirus infections usually remain latent and rarely cause symptoms in their fungal hosts; however, some mycoviruses can reduce the host’s virulence, which is called hypovirulence [8]. Hypovirulence-associated mycoviruses in plant-pathogenic fungi have attracted much attention from researchers and farmers owing to their potential biocontrol function and ability to reduce losses of forests and crops [6,9]. Mycovirus Cryphonectria hypovirus 1 (CHV1) infection is associated with attenuated virulence, reduced pigmentation, and suppressed asexual sporulation in its host [10]. CHV1 was the first mycovirus that was successfully used as a biological control agent to manage chestnut blight, caused by the plant pathogenic fungus *Cryphonectria parasitica* [11].

*Fusarium* is an important genus of plant pathogenic fungi, which causes serious damage to many field, ornamental, forest, and horticultural crops [12]. Many species of *Fusarium* are reported to be hosts of mycoviruses, including *F. andiyazi*, *F. asiaticum*, *F. boothii*, *F. circinatum*, *F. coeruleum*, *F. culmorum*, *F. globosum*, *F. graminearum*, *F. incarnatum*, *F. langsethiae*, *F. oxysporum*, *F. poae*, *F. pseudograminearum*, *F. sacchari*, *F. sibiricum*, *F. solani, F. verticillioides*, *F. virguliforme*, and so on [13,14,15,16,17]. However, only two mycoviruses are associated with *F. pseudograminearum*, Fusarium pseudograminearum megabirnavirus 1 (FpgMBV1) and Fusarium pseudograminearum mitovirus 1 (FpgMV1) [18,19]. Further research has shown that FpgMBV1 had a hypovirulence effect on *F. pseudograminearum* [20].

Mycoviruses are generally transmitted intracellularly in nature during cell division, hyphal anastomosis (horizontal transmission), and sporogenesis (vertical transmission). Their natural host ranges are limited to individuals within the same or closely related natural vegetative compatibility groups [7]. To date, a few studies have reported the presence of mycoviruses in *F. pseudograminearum*, the primary pathogen of wheat FCR, limiting the development of mycovirus-mediated biological control technology to control the FCR disease.

In the present study, we identified and characterized a novel double-stranded RNA (dsRNA) mycovirus in *F. pseudograminearum* strain WC9-2, which was shown to be an isolate of the Fusarium graminearum dsRNA mycovirus 4 (FgV4) [21], which we labeled as a FgV4-WC9-2. FgV4-WC9-2 genomic organization, and phylogenetic analyses were performed to elucidate the phylogenetic relationship. The effects of FgV4-WC9-2 on the biological characteristics of the host *F. pseudograminearum* were also investigated. This study strengthened our understanding of mycoviruses in *F. pseudograminearum*, discovered a new host for the virus, and provided theoretical bases and biocontrol resources for the controlling of wheat FCR disease.

## 2. Materials and Methods

### 2.1. Fungal Strain and Culture Conditions

Strain WC9-2 of *F. pseudograminearum*, which contains viral dsRNA, was isolated from a diseased wheat stem showing FCR symptoms in Weifang city, Shandong province, China. It was identified by the molecular method, which has been described in the report of Ma et al. [5]. The *F. pseudograminearum* strain was cultured on potato dextrose agar (PDA) plates at 25 °C in dark. Mycelial agar plugs with a diameter of 5 mm were stored in sterilized liquid paraffin at 4 °C until use.

### 2.2. DsRNA Extraction and Purification

For dsRNA extraction, the PDA plate was covered with cellophane membrane, then mycelial agar plugs (three per plate) were inoculated on the plate’s surface and cultured at 25 °C in the dark for 4 days. The mycelia of strain WC9-2 were harvested from the plate surface, ground to a fine powder using liquid nitrogen, and their dsRNA was extracted via the method previously described by Morris and Dodds [22]. The single-stranded RNA (ssRNA) and DNA contaminants in extracted dsRNA were eliminated using S1 nuclease and DNase I (RNase-free) (TaKaRa, Dalian, China), respectively. The 1.0% agarose gel electrophoresis stained with ethidium bromide (EB) was used to analyze purified dsRNA. Each dsRNA segment was separately excised under UV-induced EB fluorescence and purified by a gel extraction kit (Aidlab Biotechnologies, Beijing, China) following with the manufacturer’s instructions, and was then dissolved in 20 μL RNAase-free water and stored at –80 °C until needed.

### 2.3. Complementary DNA (cDNA) Cloning, Sequencing, and Phylogenetic Analysis

The cDNA cloning of each dsRNA segment was performed using a random primer (5′-CGATCGATCATGATGCAATGCNNNNNN-3′), as previously described [23]. The amplified PCR products were cloned into the pClone007 vector using a pClone007 versatile simple vector kit (TsingKe, Qingdao, China) and then transferred into *Escherichia coli* strain DH5α cells. Positive clones were selected for Sanger sequencing. To clone the 5′ and 3′ terminal sequences of each dsRNA segment, the method of RNA ligase-mediated rapid amplification of cDNA ends (RLM-RACE) was used, as previously described by Zhou et al. [24]. The bases at each site were confirmed by sequencing at least three independent overlapping clones in both directions.

Using the open reading frame (ORF) Finder program (https://www.ncbi.nlm.nih.gov/orffinder/ (accessed on 28 November 2024)) we analyzed the potential ORFs in each nucleotide sequence. The standard codon usages were used to find the ORF. Sequence similarity searches for nucleic acids were conducted using the BLASTn program, or sequence similarity searches for putative proteins were conducted using the BLASTp program against the National Center for Biotechnology Information (NCBI) database. Multiple alignment of amino acid (aa) sequences of the deduced RNA-dependent RNA polymerase (RdRp) domain was performed using the CLUSTAL_X program [25]. The phylogenetic tree for the aligned sequences was constructed using the neighbor-joining (NJ) method in Molecular Evolutionary Genetic Analysis (MEGA) software version 7.0, with a bootstrap of 1000 replicates [26].

### 2.4. Mycovirus Elimination from F. pseudograminearum Strain WC9-2

A ribavirin treatment of *F. pseudograminearum* strain WC9-2 was used to establish an isolate cured of mycovirus infection. Strain WC9-2 containing the mycovirus was cultured on a (PDA) plate amended with 100 µM ribavirin at 25 °C in the dark for 4 days. A hyphal tip from the edge of developing colony was transferred to a fresh PDA plate containing 100 µM ribavirin and incubated for 4 days at 25 °C in the dark. After each subculture for five generations, mycelia were collected for dsRNA extraction. The strain lacking dsRNAs was identified by electrophoresis, and the absence of dsRNAs was confirmed by reverse transcription-polymerase chain reaction (RT-PCR). The cured strain was designated as WC9-2-VF (mycovirus-free).

### 2.5. Impact of the Mycovirus on Host Biological Properties

To assess the effects of the mycovirus on biological characteristics of the fungal host, two isogenic strains, WC9-2 and WC9-2-VF, were used. Each strain was individually cultured on PDA for 4 days at 25 °C in the dark. Mycelial agar plugs (5 mm in diameter) punched from the colony margin of each strain were placed on fresh PDA plates and incubated at 25 °C in the dark for determination of the colony growth and for observation of the colony morphology. Colonial diameters were measured after culturing mycelial plugs on PDA plates for 5 days using the cross intersect method. Three mycelial plugs were inoculated into 100 mL carboxymethylcellulose sodium (CMC) fluid medium and cultured for 4 days (25 °C, 180 rpm). Then, the mycelium solution was filtered through two layers of sterile Miracloth, and the conidia suspension was diluted 10 times with sterile water. The conidia concentration was counted using a hemacytometer and the sporulation ability was calculated. The conidia size (width and length) was measured under a compound microscope (objective × 40).

The virulence of the two strains was determined following the method described by Ma et al. [5] with slight modifications. Briefly, eight days after inoculation, the area of typical browning at the crown and base of each stem was measured and recorded. The 6-point rating system modified from Smiley et al. [27] was used to evaluate the disease severity (DS), as follows: 0 = the plant is healthy without any tissue discoloration; 1 = the coleoptile is brown, and the area of browning is <25%; 2 = the coleoptile is brown, and the area of browning is 25~50%; 3 = the coleoptile is brown, and the area of browning is 51~75%; 4 = the coleoptile is brown, and the area of browning is 76~100%; 5 = the area of browning extends from the bottom to the top beyond the coleoptile. The calculation formula of disease index (DI) is DI = [100 × ∑ (n × corresponding DS)]/(N × 5), where n represents the number of the infected wheat seedlings corresponding to each disease grade, and N represents the total number of inoculated wheat seedlings.

### 2.6. Data Analysis

Microsoft Excel 2019 was used for statistical analysis, and one-way analysis of variance (ANOVA) was used at the *p* < 0.05 level.

## 3. Results

### 3.1. A Pattern of DsRNAs in F. pseudograminearum Strain WC9-2

Nucleic acids enriched in dsRNA were extracted from mycelia of *F. pseudograminearum* strain WC9-2; we eliminated ssRNA and DNA contaminants with S1 nuclease and DNase I, and then the product was analyzed by 1.0% agarose gel electrophoresis. The results show that two dsRNAs (named dsRNA1 and dsRNA2 according to the length of the nucleic acid sequence) were detected (Figure 1A). The full-length cDNA sequences of dsRNA1 and dsRNA2 were obtained by assembling partial-length cDNAs, random-primed, combined with the RLM-RACE method. The complete sequences were deposited into GenBank with the accession numbers of PQ660771 and PQ660772, respectively.

### 3.2. Genome Analysis of the FgV4-WC9-2

The complete sequences of dsRNA1 and dsRNA2 contained 2194 and 1738 base pairs (bp), respectively. Sequence analysis shows that dsRNA1 contained a single ORF (ORF1) and dsRNA2 contained two discontinuous ORFs (ORF2-1 and ORF2-2) in the positive strand. The 5′-untranslated region (UTR) and 3′-UTR of dsRNA1 were 80 bp and 86 bp long, respectively, whereas those of dsRNA2 were 119 bp and 101 bp long, respectively (Figure 1B). The ORF1 (nt 81–2108) of dsRNA1 was predicted to encode a protein of 675 amino acids (aa) and had a conserved RdRp domain. ORF2-1 (nt 120–1070) and ORF2-2 (nt 1281–1637) of dsRNA2 were found to encode putative proteins of 316 aa and 118 aa (Figure 1B).

The BLASTn search showed that the nucleic acid sequence of dsRNA1 had identities of 96.06%, 82.60%, 96.00%, 95.85%, and 89.11% to those of Fusarium graminearum dsRNA mycovirus 4 (FgV4), Fusarium graminearum dsRNA mycovirus 5 (FgV5), Fusarium pseudograminearum orthocurvulavirus 1 (FpgOV1), Fusarium pseudograminearum orthocurvulavirus 2 (FpgOV2), and Fusarium pseudograminearum orthocurvulavirus 3 (FpgOV3), respectively. The sequence of dsRNA2 had identities of 87.80% and 80.16% with the counterparts of FgV4 and FgV5, respectively (Table 1). The identities of the 5′-UTR of the virus with those of FgV4 and FgV5 ranged from 8.42% to 85.83%. The 3′-UTR of the virus had identities ranging from 87.25% to 98.84% with those of FgV4 and FgV5 (Table 1).

The BLASTp analysis of the aa sequence of ORF1 revealed identities of 96.16%, 96.42%, 99.39%, 99.44%, and 98.82% to those of FgV4, FgV5, FpgOV1, FpgOV2, and FpgOV3, respectively (Table 2). Therefore, all these viruses belong to the same species, according to the ICTV species demarcation criteria for the genus *Orthocurvulavirus*, proposed as 85% identity. The sequence of ORF2-1 was similar to its counterparts in FgV4 and FgV5, with 91.26% and 91.77% identities, respectively. The sequence of ORF2-2 showed 95.76% and 74.58% identity to those of FgV4 and FgV5, respectively (Table 2). Overall, these results suggest that dsRNA1 and dsRNA2 are the genomic components of a novel isolate of FgV4, designated as FgV4-WC9-2.

### 3.3. Phylogenetic Analysis of FgV4-WC9-2

A phylogenetic tree was established on the basis of RdRp domains of a virus described in this work and previously reported members of the families *Chrysoviridae*, *Totiviridae*, *Amalgaviridae*, *Curvulaviridae*, and *Partitiviridae*. The result show that the virus had the closest relationship with FgV4 and FpgOV1, forming a clade with a bootstrap of 88%, and then clustered with viruses in *Orthocurvulavirus* and *Curvulaviridae* (Figure 2). This result confirms that this virus was a novel isolate of Fusarium graminearum dsRNA mycovirus 4.

### 3.4. Impact of FgV4-WC9-2 on Host Biological Properties

To determine the biological effects of FgV4-WC9-2 on its host, FgV4-WC9-2 was eliminated through the ribavirin treatment method. The FgV4-WC9-2-cured strain, named as WC9-2-VF, as well as FgV4-WC9-2-infected strain WC9-2, were used for the biological property assays.

After dark culturing on PDA plates at 25 °C for 5 days, there was no significant difference in colony morphology (including mycelial texture and pigment color) between the FgV4-WC9-2-cured strain WC9-2-VF and the original FgV4-WC9-2-infected strain WC9-2 (Figure 3A). However, the colony diameter of the FgV4-WC9-2-infected strain WC9-2 was significantly smaller than that of the FgV4-WC9-2-cured strain WC9-2-VF (Figure 3A). The colony diameters of the two strains were measured using the cross intersect method. The average mycelial growth rate of WC9-2 was 12.3 mm/day, which was significantly (*p* < 0.05) slower than that of the FgV4-WC9-2-cured strain WC9-2-VF (16.5 mm/day) (Figure 3B).

The sporulation ability, expressed as the number of conidia per milliliter, was determined at the fourth day post incubation. The concentration of conidia suspension for FgV4-WC9-2-infected strain WC9-2 was 3.56 × 10^6^ conidia/mL, which is not significantly (*p* < 0.05) lower than that of the FgV4-WC9-2-cured strain WC9-2-VF (3.78 × 10^6^ conidia/mL) (Figure 3C). The average conidia length of FgV4-WC9-2-infected strain WC9-2 was 37.73 µm, while that of the FgV4-WC9-2-cured strain WC9-2-VF was 33.91 µm (Figure 3D). The average conidia width of the FgV4-WC9-2-infected strain WC9-2 and that of the FgV4-WC9-2-cured strain WC9-2-VF were 3.46 µm and 3.47 µm, respectively (Figure 3E). There were no significant differences in average conidia length and width between these two strains.

The influence of FgV4-WC9-2 on the pathogenicity of *F. pseudograminearum* was investigated. Both the FgV4-WC9-2-infected strain WC9-2 and the FgV4-WC9-2-cured strain WC9-2-VF were inoculated at the base of each wheat seedling stem. The two strains produced dark brown lesions on the wheat seedling coleoptile, and sizes of the lesions induced by strain WC9-2-VF were a little bigger than those incited by strain WC9-2 (Figure 4A). The disease incidence of wheat seedlings caused by strain WC9-2 was 86.67%, while strain WC9-2-VF was 93.33% (Figure 4B). The disease indices of wheat seedlings caused by strain WC9-2 and strain WC9-2-VF were 26.00 and 30.00, respectively (Figure 4C). There were no significant differences in disease incidence and disease index between these two strains.

## 4. Discussion

In this study, we identified and characterized a dsRNA mycovirus from *F. pseudograminearum* strain WC9-2, which was isolated from a diseased wheat sample showing typical FCR symptoms. Based on the homology BLASTn and BLASTp searches, genome organization comparison, and phylogenetic analysis, we propose that this dsRNA mycovirus is a novel isolate of FgV4, which belongs to the genus *Orthocurvulavirus*, family *Orthocurvulaviridae*, and is designated as FgV4-WC9-2.

Some mycoviruses have the potential to be biological agents for controlling fungal diseases owing to their ability to cause hypovirulence on their host’s phytopathogenic fungi [6,7,9,28,29,30,31,32,33,34]. To date, several members in the recently established genus *Orthocurvulavirus* within the family *Orthocurvulaviridae* have been reported. For example, FgV4 and FgV5 have been identified from *Fusarium graminearum*, an important plant pathogen that causes head and seedling blight [21,35], Heterobasidion RNA virus 6 (HetRV6) was reported to infect conifer root-rot fungus *Heterobasidion annosum* sensu lato [36], Rhizoctonia solani dsRNA virus 1 (RsRV1) was found in *Rhizoctonia solani*, the causal agent of rice sheath blight [37], Gremmeniella abietina RNA virus 6 (GaRV6) was identified in the European race of *Gremmeniella abietina*, the causal agent of stem canker and shoot blight on numerous conifers [38], Trichoderma harzianum bipartite mycovirus 1 (ThBMV1) was identified in *Trichoderma harzianum* strain 137 isolated in Xinjiang province, China [39], Alternaria longipes dsRNA virus 1 (AldRV1) was isolated and characterized in *Alternaria longipes*, the pathogen of pear black spot disease [40], and Beauveria bassiana orthocurvulavirus 1 (BbOCuV1) was found in *Beauveria bassiana* Vuillemin, an entomopathogenic fungus that has been developed as a biological insecticide [41]. However, only a few members have been further studied; for example, it was found that FgV4 infection did not cause any phenotypic changes but decreased the expressions of genes associated with ribosome assembly and RNA processing [42], while HetRV6 appeared to be slightly mutualistic or cryptic to its fungal host, and its RdRp could follow a primer-independent process to initiate RNA synthesis. It was also applicable for use in molecular biology, biotechnological applications, and biomedicine [43]. It is necessary to strengthen the research on the biological properties of orthocurvulaviruses to lay a theoretical foundation for the application of biocontrol.

*F. pseudograminearum* is the predominant species causing wheat FCR disease in many wheat-planting regions [3,4,5]. The use of fungicides is the main method used in controlling wheat FCR disease, but it causes environmental problems and is labor-intensive. Therefore, environmentally friendly and highly efficient alternative methods are desirable, such as hypovirulence mediated by the mycovirus in plant pathogenic fungi [6]. Up to now, only two mycoviruses isolated from *F. pseudograminearum* have been reported, FpgMBV1 and FpgMV1 [18,19]. Sequences of three orthocurvulaviruses isolated from *F. pseudograminearum* were searched in the GenBank database, and the three orthocurvulaviruses were designated as FpgOV1, FpgOV2, and FpgOV3, respectively. In spite of the different names assigned to these viruses, our study revealed that all of them belong to the same virus species, along with FgV4, as they mutually share amino acid identities exceeding the proposed species threshold of less than 85% identical residues. FpgMBV1 had a hypovirulence effect on *F. pseudograminearum* [20]. Our study isolated and characterized a novel isolate of FgV4 (FgV4-WC9-2) from *F. pseudograminearum*. FgV4-WC9-2 infection did not change the colony morphology, but it was shown to significantly decrease colony growth rate. Infection with FgV4-WC9-2 could reduce sporulation ability, change the conidia size and reduce the pathogenicity of the host to a certain extent. These results imply that FgV4-WC9-2 might have the potential to be used as a biological control agent against *F. pseudograminearum*.

This work systematically studied the effects of FgV4-WC9-2 on its host *F. pseudograminearum*, offering the first report of the biological effects of orthocurvulavirus on *F. pseudograminearum* in the world. The exploitation of identification, molecular characterization, genetic evolution, and hypovirulent characteristics of mycovirus resources is the key prerequisite for the efficient application of a mycovirus in biological control. Thus, it is particularly important to excavate mycoviral resources from *F. pseudograminearum* strains in order to accumulate candidate materials for the biocontrol of wheat FCR disease.

## Figures and Tables

**Figure 1 microorganisms-13-00418-f001:**
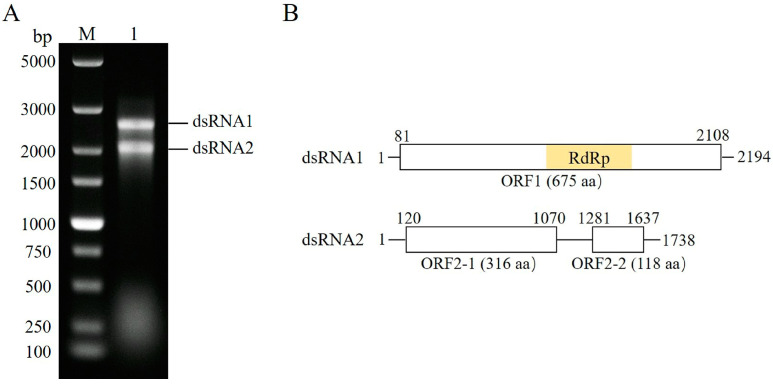
Analysis of the two FgV4-WC9-2 dsRNA segments. (**A**) The dsRNAs from the *F. pseudograminearum* strain WC9-2 were analyzed using 1.0% agarose gel. Lane M, DL 5,000 DNA marker; Lane 1, dsRNAs extracted from strain WC9-2 and treated with S1 nuclease and DNase I. (**B**) The genome organization of FgV4-WC9-2 is shown with key nucleotides and amino acids. The RNA-dependent RNA polymerase (RdRp) domain is indicated. The untranslated regions (UTRs) and open reading frames (ORFs) are indicated with lines and boxes, respectively.

**Figure 2 microorganisms-13-00418-f002:**
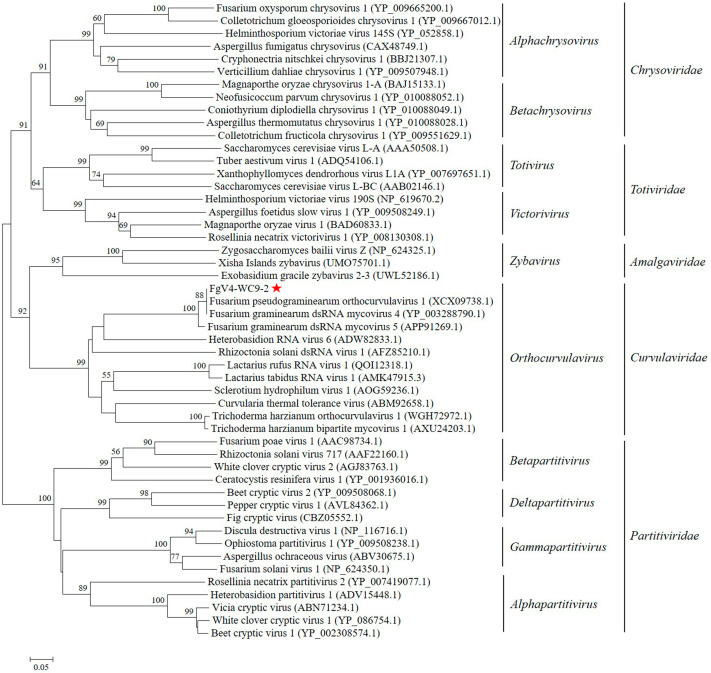
Phylogenetic analysis of the RdRp domains of FgV4-WC9-2 and other selected dsRNA viruses within the families *Chrysoviridae*, *Totiviridae*, *Amalgaviridae*, *Curvulaviridae*, and *Partitiviridae* using the neighbor-joining (NJ) method with 1000 bootstrap replicates in MEGA 7.0. We display bootstrap values greater than 50% above the branch. The scale bar indicates a genetic distance of 0.05. The location of FgV4-WC9-2 is indicated by the red star.

**Figure 3 microorganisms-13-00418-f003:**
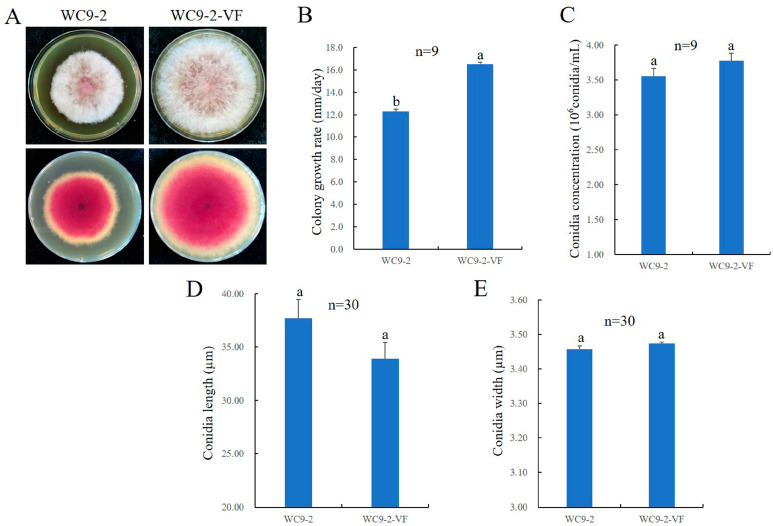
Effects of FgV4-WC9-2 on host morphology, growth, sporulation ability, and conidia size. (**A**) Colony morphologies of FgV4-WC9-2-infected strain WC9-2 and FgV4-WC9-2-cured strain WC9-2-VF, which were dark cultured on PDA plates for 5 days at 25 °C. (**B**) Radial mycelial growth rates on PDA (25 °C) of strain WC9-2 and WC9-2-VF, respectively. (**C**) Sporulation abilities in CMC (25 °C, 180 rpm) of strain WC9-2 and WC9-2-VF, respectively. (**D**,**E**) Conidia lengths (**D**) and conidia widths (**E**) of strains WC9-2 and WC9-2-VF, respectively. Microsoft Excel 2019 was used for the one-way ANOVA statistical analyses at *p* < 0.05 level. Significant difference is indicated by different letters. No significant difference was indicated by the same letters.

**Figure 4 microorganisms-13-00418-f004:**
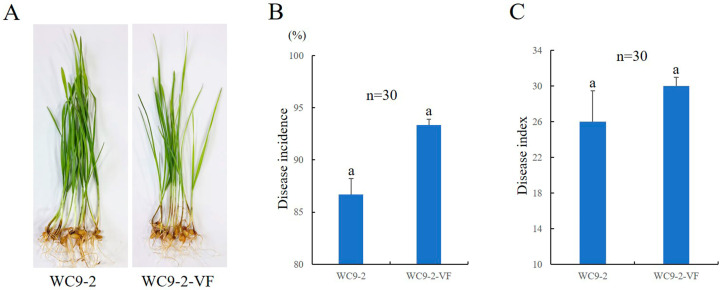
Effect of FgV4-WC9-2 on host pathogenicity. (**A**) Representative symptoms in wheat seedlings (cultivar ‘Jimai 22’) following inoculation with FgV4-WC9-2-infected strain WC9-2 and FgV4-WC9-2-cured strain WC9-2-VF at 8 dpi. (**B**) Disease incidence of wheat seedlings inoculated with strain WC9-2 and WC9-2-VF, respectively. (**C**) Disease index of wheat seedlings inoculated with strain WC9-2 and WC9-2-VF, respectively. Microsoft Excel 2019 was used for the one-way ANOVA statistical analyses at the *p* < 0.05 level. No significant difference is indicated by the use of the same letters.

**Table 1 microorganisms-13-00418-t001:** Analyses of nucleic acid sequence among similar orthocurvulaviruses.

	dsRNAs (bp/Identity)		5′-UTR (bp/Identity)		3′-UTR (bp/Identity)		Complete Sequence
	dsRNA1	dsRNA2	dsRNA1	dsRNA2	dsRNA1	dsRNA2	
FgV4	2383/96.06%	1739/87.80%	158/8.42%	120/85.83%	86/98.84%	102/97.06%	yes
FgV5	2030/82.60%	1740/80.16%	99/21.21%	120/68.33%	86/91.86%	102/87.25%	yes
FpgOV1	2076/96.00%	- ^a^	-	-	-	-	no
FpgOV2	1205/95.85%	-	-	-	-	-	no
FpgOV3	777/89.11%	-	-	-	-	-	no

^a^ No relevant information was found.

**Table 2 microorganisms-13-00418-t002:** Analyses of amino acid sequences among similar orthocurvulaviruses.

	ORF1 (aa/Identity)	ORF2-1 (aa/Identity)	ORF2-2 (aa/Identity)
FgV4	712/96.16%	338/91.26%	118/95.76%
FgV5	614/96.42%	316/91.77%	118/74.58%
FpgOV1	651/99.39%	- ^b^	-
FpgOV2	359/99.44%	-	-
FpgOV3	254/98.82%	-	-

^b^ No relevant information was found.

## Data Availability

The genomic sequences of FgV4-WC9-2 have been deposited in GenBank with the following accession numbers: PQ660771 and PQ660772.

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
