# Peer review of "Molecular and Biological Characterization of an Isolate of Fusarium graminearum dsRNA mycovirus 4 (FgV4) from a New Host Fusarium pseudograminearum"

_microorganisms, 2025, doi:10.3390/microorganisms13020418_

Round 1

Reviewer 1 Report

Comments and Suggestions for Authors

Dear authors, the research presented "A Novel Orthocurvulavirus Induces Hypovirulence in Phytopathogenic Fungus Fusarium pseudograminearum" is important in the current context of identifying new biocontrol possibilities. 

The use of mycoviruses a s biocontrol agents represent a step forward for the development of this field.

Keywords - some of the keywords should be changed with other similar ones or different to no repeat the terms from the title. In this way, the keywords will complete the image presented in the title.

Introduction section - The information presented within lines 37-42 and lines and 56-64 complete well each other.

In this section the research problem is well introduced and the importance of the study is stated.

The aim of the research is clearly explained.

The results are clear and present all the parameters necessary to explain the changes induced by virus to the targeted pathogen. Each figure and table is explained well.

 Discussion section - the authors compare their findings with other international references in the field. Comparison are made with other fungal species, which enlarge the application of the mechanism observed by the authors, and expand the knowledge related to the pathogen mycoviruses. 

The future perspectives are explained in the last paragraph, and the application of the results in new biological control techniques is presented.

Author Response

Dear Reviewer,

  Thank you for your comments on my manuscript entitled “A Novel Orthocurvulavirus Induces Hypovirulence in Phytopathogenic Fungus Fusarium pseudograminearum” (microorganisms-3440561). The comments are very helpful for revising and improving the manuscript. I have read through the comments carefully and revised the manuscript based on these comments. Changes in the manuscript have been highlighted within the document by using red-colored text. Please evaluate the revised manuscript again. Point-by-point responses to all the comments made by you are provided.

  1. Comment 1: Dear authors, the research presented “A Novel Orthocurvulavirus Induces Hypovirulence in Phytopathogenic Fungus Fusarium pseudograminearum” is important in the current context of identifying new biocontrol possibilities.

     My response: Thank you very much for your recognition.

  1. Comment 2: The use of mycoviruses a s biocontrol agents represent a step forward for the development of this field.

     My response: Thanks for your recognition and support.

  1. Comment 3: Keywords - some of the keywords should be changed with other similar ones or different to no repeat the terms from the title. In this way, the keywords will complete the image presented in the title.

     My response: Thanks for your good suggestion. The keywords “Fusarium crown rot; Wheat; Fusarium pseudograminearum; Mycovirus; Biological characteristics” have been changed to “Fusarium pseudograminearum; Mycovirus; Orthocurvulavirus; Biological characteristics; Biocontrol” in the revised manuscript. (Lines 27-28). Hoping our answer will be approved by you. Thank you once again.

  1. Comment 4: Introduction section - The information presented within lines 37-42 and lines and 56-64 complete well each other. In this section the research problem is well introduced and the importance of the study is stated. The aim of the research is clearly explained.

    My response: Thank you very much for your recognition.

  1. Comment 5: The results are clear and present all the parameters necessary to explain the changes induced by virus to the targeted pathogen. Each figure and table is explained well.

     My response: Thank you very much for your recognition.

  1. Comment 6: Discussion section - the authors compare their findings with other international references in the field. Comparison are made with other fungal species, which enlarge the application of the mechanism observed by the authors, and expand the knowledge related to the pathogen mycoviruses.

     My response: Thank you very much for your recognition.

  1. Comment 7: The future perspectives are explained in the last paragraph, and the application of the results in new biological control techniques is presented.

    My response: Thank you very much for your recognition.

  We appreciate the reviewer’s thorough examination of our manuscript, and hope that the revised manuscript will meet with your approval. We look forward to hearing from you regarding the revision. We would be glad to respond to any further questions and comments on the revised manuscript. Once again, thank you very much for your comments and recognition.

Best regards.

Yours sincerely,

Guoping Ma

Institute of Plant Protection, Shandong Academy of Agricultural Sciences, Shandong Key Laboratory for Green Prevention and Control of Agricultural Pests

No.23788 Gongye North Road, Licheng District, Jinan 250100, P. R. China

Tel: +86-531-66658212; Email: guopingmasaas@163.com

Reviewer 2 Report

Comments and Suggestions for Authors

Authors concentrated on  hypovirulence-associated orthocurvulavirus infecting F. pseudograminearum;

Introduction is clearly presented and gives the reader sufficient information to analyze Authors obtained results, but the highlighted aim of studies should be defined.

Materials and methods were presented in detail in a repetitive way.

The strength part of these manuscript is detailed genome analyses and phylogenetic studies of novel mycovirus.

Only one subparts of results raises my reservations. The “effect of FpgOV4 on host pathogenicity” is too weak and only one analyses potentially identified pathogenicity aspects- deeper as well as more precise molecular and/or structural studies should be added. Besides of them presented symptoms on seedlings are almost indistinguishable – Please , make in molecular analyse as much as it will be possible to strengthened that point of view, because in plant-microbe interaction only disease incidence test is too weak.

On the other side, discussion part is maybe short, but it is well constructed without redundance effect.

Author Response

Dear Reviewer,

  Thank you for your comments on my manuscript entitled “A Novel Orthocurvulavirus Induces Hypovirulence in Phytopathogenic Fungus Fusarium pseudograminearum” (microorganisms-3440561). The comments are very helpful for revising and improving the manuscript. I have read through the comments carefully and revised the manuscript based on these comments. Changes in the manuscript have been highlighted within the document by using red-colored text. Please evaluate the revised manuscript again. Point-by-point responses to all the comments made by you are provided.

  1. Comment 1: Authors concentrated on hypovirulence-associated orthocurvulavirus infecting F. pseudograminearum. Introduction is clearly presented and gives the reader sufficient information to analyze authors obtained results, but the highlighted aim of studies should be defined.

My response: Thanks for your good suggestion. The highlighted aim has been added as “This study strengthened the understanding of mycoviruses in F. pseudograminearum, enriched the diversity of mycoviruses, and provided theoretical basis and biocontrol resource for the controlling of wheat FCR disease.” in the revised manuscript. (Lines 77-79). Hoping our answer will be approved by you. Thank you once again.

  1. Comment 2: Materials and methods were presented in detail in a repetitive way. The strength part of these manuscript is detailed genome analyses and phylogenetic studies of novel mycovirus.

My response: Thank you very much for your recognition.

  1. Comment 3: Only one subparts of results raises my reservations. The “effect of FpgOV4 on host pathogenicity” is too weak and only one analyses potentially identified pathogenicity aspects - deeper as well as more precise molecular and/or structural studies should be added. Besides of them presented symptoms on seedlings are almost indistinguishable – Please , make in molecular analyse as much as it will be possible to strengthened that point of view, because in plant-microbe interaction only disease incidence test is too weak.

My response: Thanks for your good suggestion. We evaluate the effect of FpgOV4 on host pathogenicity by referring to the methods described by Wang et al. (2022), Suharto et al. (2022), and Ma et al. (2024). We used method of disease incidence and disease index to evaluate fungi pathogenicity, which has been recognized by many researchers. We also agree with your opinion that precise molecular and/or structural studies will strengthen the result of effect of FpgOV4 on host pathogenicity. We will conduct this research in future studies. Hoping our answer will be approved by you. Thank you once again.

References:

Wang, J.; Li, C.; Song, P.; Qiu, R.; Song, R.; Li, X.; Ni, Y.; Zhao, H.; Liu, H.; Li, S. Molecular and biological characterization of the first mymonavirus identified in Fusarium oxysporum. Front. Microbiol. 2022, 13, 870204.

Suharto, A.R.; Jirakkakul, J.; Eusebio-Cope, A.; Salaipeth, L. Hypovirulence of Colletotrichum gloesporioides associated with dsRNA mycovirus isolated from a mango orchard in Thailand. Viruses 2022, 14, 1921.

Ma, G.; Wang, H.; Qi, K.; Ma, L.; Zhang, B.; Zhang, Y.; Jiang, H.; Wu, X.; Qi, J. Isolation, characterization, and pathogenicity of Fusarium species causing crown rot of wheat. Front. Microbiol. 2024, 15, 1405115.

  1. Comment 4: On the other side, discussion part is maybe short, but it is well constructed without redundance effect.

My response: Thanks for your recognition and good suggestion. Considering the structure of this paper, we discussed the purpose, results, significance, and prospect of this study. We accept your suggestion and will focus on the writing of the discussion section in future studies. Hoping our answer will be approved by you. Thank you once again.

We appreciate the reviewer’s thorough examination of our manuscript, and hope that the revised manuscript will meet with your approval. We look forward to hearing from you regarding the revision. We would be glad to respond to any further questions and comments on the revised manuscript. Once again, thank you very much for your comments and recognition.

Best regards.

Yours sincerely,

Guoping Ma

Institute of Plant Protection, Shandong Academy of Agricultural Sciences, Shandong Key Laboratory for Green Prevention and Control of Agricultural Pests

No.23788 Gongye North Road, Licheng District, Jinan 250100, P. R. China

Tel: +86-531-66658212; Email: guopingmasaas@163.com
